# Evaluation of Antimicrobial and Anti-Biofilm Formation Activities of Novel Poly(vinyl alcohol) Hydrogels Reinforced with Crosslinked Chitosan and Silver Nano-Particles

**DOI:** 10.3390/polym14081619

**Published:** 2022-04-16

**Authors:** Reem T. Alfuraydi, Fahad M. Alminderej, Nadia A. Mohamed

**Affiliations:** 1Department of Chemistry, College of Science, Qassim University, Buraidah 51452, Saudi Arabia; 2Department of Chemistry, Faculty of Science, Cairo University, Giza 12613, Egypt

**Keywords:** PVA/chitosan hydrogels, trimellitic anhydride isothiocyanate crosslinker, silver nanoparticles, antimicrobial activity, anti-biofilm formation activity, cytotoxicity

## Abstract

Novel hydrogels were prepared by blending chitosan and poly(vinyl alcohol), PVA, then crosslinking the resulting blends using trimellitic anhydride isothiocyanate at a concentration based on chitosan content in the blends. The weight ratios of chitosan: PVA in the blends were 1:3, 1:1, and 3:1 to produce three hydrogels symbolized as H_13_, H_11_, and H_31_, respectively. For a comparison, H_10_ was also prepared by crosslinking pure chitosan with trimellitic anhydride isothiocyanate. For further modification, three H_31_/silver nanocomposites (AgNPs) were synthesized using three different concentrations of silver nitrate to obtain H_31_/AgNPs1%, H_31_/AgNPs3% and H_31_/AgNPs5%. The structures of the prepared samples were emphasized using various analytical techniques. PVA has no inhibition activity against the tested microbes and biofilms. The antimicrobial and anti-biofilm formation activities of the investigated samples was arranged as: H_31_/AgNPs5% ≥ H_31_/AgNPs3% > H_31_/AgNPs1% > H_10_ > H_31_ > H_11_ > H_13_ > chitosan. H_31_/AgNPs5% and H_31_/AgNPs3% were more potent than *Vancomycin* and *Amphotericin B* against most of the tested microbes. Interestingly, H_31_ and H_31_/AgNPs3% were safe on the normal human cells. Consequently, hydrogels resulting from crosslinked blends of chitosan and PVA loaded with AgNPs in the same structure have significantly reinforced the antimicrobial and inhibition activity against the biofilms of PVA.

## 1. Introduction

The biofilms are usually made up of complicated groups of living microorganisms bound to a layer and surrounded by a self-producing extracellular matrix that consists of polysaccharides, extracellular DNA, and proteins. Thus, the biofilms problem increases the complicating state of resistance of the antimicrobial agents. It is well established that a great percent of the infections of human includes biofilms. The bacteria that form biofilms are much more resistant to antibiotics. The most known pathogens, including protozoa, bacteria, viruses, fungi, and pathogenic cyanobacteria, are involved in the formation of biofilms. The latter depends on the kind of microorganism and the matter. Biofilms showed an ability for trapping particles, including diversified minerals and ingredients of the host system, such as platelets, fibrin, and red blood cells. Biofilms are composed of many cells of microbes and bind with surfaces such as medical equipment, water pipe systems, and biological tissues [1].

The microorganisms that form biofilms are implicated in several infected diseases such as periodontitis, dental caries, otitis media, and osteomyelitis and in chronic diseases such as pulmonary infections of cystic fibrosis patients [1]. The resistance of several strains of microorganisms to the traditional antibiotic has directed the interest towards searching for new alternating materials as efficient antimicrobial agents that act with different mechanisms than that of conventional antibiotics. Most hospital infections are greatly related to potent biofilm-producing *Pseudomonas aeruginosa* and *Staphylococcus aureus* [2]. It was found that the biofilms of *Candida* species have a high resistance to antifungal agents and the host immune system than free cells, resulting in an elevated mortality rate [3]. Thus, anti-biofilm activity should be assessed for its ability to overcome antimicrobial resistance.

PVA, a synthetic eco-friendly vinyl polymer, is water-soluble, hydrophilic, chemically stable, inherently non-toxic, non-carcinogenic, bio-compatible, elastic, and able to form gels and films [4]. It is applied as a basic material in several biomedical fields in artificial cartilages, replacement of skin material, contact lenses, and reconstruction of the vocal cords [5]. However, it has no activity against microorganisms such as bacteria and fungi because it has a weakly acidic nature due to its hydroxyl groups, so it cannot interact with the negatively charged membranes of bacteria and fungi [6,7]. Thus, the development of its antimicrobial characteristics represents a key to further expanding its application domains. Its combination with one of the natural antimicrobial biopolymers, such as chitosan, is considered an efficient approach to improve its antimicrobial activity.

Chitosan, the deacetylated state of naturally abundant chitin, is a copolymer of N-acetyl-D-glucosamine and D-glucosamine. Chitosan has received immense attention due to its fascinating properties, such as antimicrobial activity, high viscosity, polyelectrolyte behavior, mineral, fat and dyes binding properties, biodegradability, film-forming capability, hypolipidemic activity, antioxidant property, and accelerating the healing of wounds. These properties allow chitosan to be used in a wide range of applications in both medicinal and environmental domains [8,9,10,11]. Chitosan is insoluble in water and even in organic solvents, while it can be dissolved in aqueous acidic solutions. The amino groups of chitosan cannot be protonated in both neutral and alkaline mediums, and subsequently, it is not soluble in water. In contrast, in the acidic medium, it forms the soluble protonated polysaccharide form. Different chemical modifications of chitosan can alter its properties and make it a wonderful choice for various applications in many fields [12,13,14].

The research field of polymeric hydrogels has attracted an increasing amount of attention due to the growth of their wonderful applications in pharmaceutical and biomedical applications [15]. The hydrogels applied in the medical domain are usually prepared using natural biopolymers due to their bio-compatibility, bio-degradability, and non-toxicity. The polymeric chains network in hydrogels can be constructed by physical, chemical, or radiation crosslinking [16]. Hydrogels can imbibe considerable quantities of water or biological fluids and maintain their featured three-dimensional structure without dissolution [17]. Hydrogels have a sensitivity toward the parameters of the environment, such as temperature, pH, the concentration of the solvent, ionic concentration, and electric fields [18]. One of these characteristics or more resulted in the use of the hydrogels in a wide variety of applications as drug delivery systems, tissue engineering culture, cell scaffolds, and antimicrobial agents [19]. 

Several researchers made PVA-based hydrogel systems via crosslinking either physically [6,7], chemically [4], or by UV radiation [20] to reinforce the antimicrobial activity of PVA for biomedical applications and for their use in drug delivery systems. Further, PVA membranes modified with hydroxyapatite-silver phosphate nanoparticles that were produced by supercritical carbon dioxide phase inversion [21] and silver-based cellulose/PVA materials [22] have shown excellent antibacterial activities. Recently, it has been reported that *O*-chitosan quaternary ammonium salt/PVA/graphene oxide could be used as a dual self-healing bacteriostatic hydrogel since it has high potency to inhibit *E. coli* and *S. aureus* and displayed a good behavior to release the serum albumin of bovine [23]. Moreover, PVA/*N*-succinyl chitosan/lincomycin composite hydrogels showed antibacterial activities and could be used for wound dressing [24]. PVA and p-coumaric acid-modified water—soluble chitosan blended membrane has shown a water-induced shape memory behavior [25].

Silver nanoparticles, AgNPs, have proven worthiness in inhibiting the activity of various types of pathogenic microorganisms [26]. They could be utilized as anticancer, antifungal, antibacterial, wound dressing, skin ointments and creams for healing burns, and in catheters [4,14]. In the last few years, the green synthesis of AgNPs has attracted a great deal of attention because of its low cost and eco-friendly nature. This is achieved by using a solvent known for its non-toxicity, similar to water, and safe, natural materials known for their reducing character similar to natural polysaccharides, including chitosan, starch, xanthan gum, and agar, via their -OH groups [27,28]. Additionally, chitosan is a strong ligand for chelation with metal ions; thus, the electron-rich NH_2_ groups of chitosan can effectively electrostatically interact with silver ions, leading to an increment in the stability and consequently in the dispersion of AgNPs [29].

In our previous work, we have crosslinked parent chitosan using trimellitic anhydride isothiocyanate through the reaction of both the anhydride and isothiocyanate groups with the amino groups of chitosan, resulting in the incorporation of the bioactive thiourea and amide linkages in addition to carboxylic groups into chitosan. This considerably enhanced both the antimicrobial activity and the adsorption capacity for anionic Congo red and cationic Basic Red 12 dyes of the produced crosslinked chitosan hydrogels [30,31,32]. However, the PVA-based hydrogels incorporating chitosan crosslinked with trimellitic anhydride isothiocyanate were not previously prepared. 

The present study is concerned with the preparation of novel cross-linked hydrogels composed of chitosan and PVA in various weight ratios. Trimellitic anhydride isothiocyanate was used as a chemical crosslinking agent at a concentration based on the chitosan content in the hydrogel. A hydrogel composed of only crosslinked chitosan was prepared to be used for comparison. Moreover, some hydrogels impregnated with AgNPs into their matrices were also prepared. The structure of the hydrogels and silver nano bio-composites was checked using elemental analyses, Fourier-transform infrared spectroscopy (FTIR), X-ray photoelectron spectroscopy (XPS), X-ray diffraction (XRD), Energy-dispersive X-ray spectroscopy (EDS), Scanning electron microscopy (SEM) and Transmission electron microscopy (TEM) measurements. The activity of these hydrogels and silver nano bio-composites against various strains of Gram-positive bacteria, Gram-negative bacteria, and the fungi was inspected. The study was extended to evaluate their activity against biofilm formation and cytotoxicity.

## 2. Materials and Methods

### 2.1. Materials

Chitosan was purchased from Funakoshi Co. Ltd. (Tokyo, Japan). Its degree of deacetylation and molecular mass are 88.2% and 2.9–3.1 × 10^5^ g mol^−1^, respectively. Ammonium thiocyanate, polyethylene glycol-400, and poly(vinyl alcohol) (PVA, code: 341584) were obtained from Sigma–Aldrich (Saint Louis, MO, USA). The degree of hydrolysis and molecular mass of PVA are 99.0–99.8% and 8.9–9.8 ×10^4^ g mol^−1^, respectively. 2,3-Bis(2-methoxy-4-nitro-5-sulfophenyl)-2*H*-tetrazolium-5-carboxanilide inner salt (XTT sodium salt) (≥90%) and 3-(4,5-dimethyl-2-thiazolyl)-2,5-diphenyl-2H-tetrazolium bromide (MTT) (98%) were supplied by Sigma (Saint Louis, MO, USA). Trimellitic anhydride chloride, the other used chemicals and solvents, were extra pure and received from Aldrich (Hamburg, Germany).

### 2.2. Methods

#### 2.2.1. Synthesis of Trimellitic Anhydride Isothiocyanate Crosslinker

A precalculated molar amount of trimellitic anhydride chloride was slowly added to an equivalent molar amount of ammonium thiocyanate dissolved in 30 mL of dichloromethane. To this reaction mixture, 1 mL of a catalyst for phase transfer (polyethylene glycol-400) was added and stirred for 2 h at 25 °C. The by-product (a white precipitate, ammonium chloride) was isolated by filtration. The trimellitic anhydride isothiocyanate crosslinker (the filtrate, Figure 1) was produced as reported in our previous work [30].

#### 2.2.2. Synthesis of Chitosan/PVA Hydrogels 

Chitosan was stirred in aqueous acetic acid solution (1% *v*/*v*) at 25 °C overnight for complete dissolution to obtain chitosan solution (1.5 wt% *v*/*v*). PVA was stirred in distilled water at 60 °C for 3 h till complete dissolution to produce PVA solution (5.0 wt% *v*/*v*). PVA solution was added to chitosan solution and stirred well together in a water bath at 60 °C for 1 h. Then, trimellitic anhydride isothiocyanate crosslinker (Figure 1) was gradually added to the resulting chitosan/PVA blend solutions of various weight ratios (Table 1), stirred well at 60 °C for 2 h, then at 25 °C overnight. The molar ratio of chitosan component to trimellitic anhydride isothiocyanate crosslinker was 2:1, respectively (Table 1 and Figure 2). To remove the residual acetic acid medium, the resulting homogenous cross-linked hydrogels were neutralized using sodium bicarbonate solution until reaching pH 7, soaked in methanol overnight to dewater and desalt, filtered, and dried at 60 °C until reaching a constant weight. The colour of the produced hydrogels ranged from pale yellow to deep yellow based on the crosslinker content that depended on the chitosan content in the hydrogel (Appendix A). The reactants ratio was listed in Table 1 to obtain four hydrogels denoted as H_10_, H_11_, H_13_, and H_31_.

#### 2.2.3. Synthesis of H_31_/Silver Nanoparticle (H_31_/AgNP) Composites 

H_31_/AgNP composites were synthesized according to the procedure described by Hebeish et al. [33]. Three various precalculated amounts of AgNO_3_ were individually stirred in 10 mL of deionized H_2_O. Each AgNO_3_ solution was added to a fixed weight of H_31_ (1 g) that was swollen in 50 mL of trisodium citrate solution of 0.01 mol L^−1^ concentration, stirred well at 25 °C for 24 h. An appreciable change in the color of the three resulting composites was observed that ranged from faint brown to deep brown in proportion to the increment of AgNO_3_ concentrations. This referred to the reduction of Ag^+^ cations to AgNPs inside the matrices of H_31_ (Figure 3). The three H_31_/AgNP composites were filtered, washed frequently with deionized H_2_O, then methanol, and dried at 60 °C. The AgNO_3_ concentrations were 1 wt%, 3 wt% and 5 wt% on the basis of H_31_, obtaining three nano-biocomposites denoted as H_31_/AgNPs1%, H_31_/AgNPs3%, and H_31_/AgNPs5%, respectively.

#### 2.2.4. Solubility Study

To determine the soluble fractions, 50 mg of chitosan, PVA, H_10_, H_13_, H_11_, H_31_, and H_31_/AgNP composites were stirred individually at 60 °C overnight in 50 mL of different solvents such as water, 1% aqueous solution of acetic acid, N,N-dimethyl formamide, tetrahydrofuran, chloroform, ethanol, acetone, and methylene chloride. Then, the residual samples were dried at 60 °C to a fixed weight. The soluble fraction could be calculated using Equation (1).
% Soluble fraction = [(W_0_ – W_1_)/W_0_] ×100(1)
where W_0_ and W_1_ are the initial weight and the dried weight of the sample, respectively.

#### 2.2.5. Minimum Inhibitory Concentration (MIC) Assay 

MIC is the least concentration that completely inhibits microbial growth. MIC values were determined using the micro-dilution method [34]. The preparation of the suspensions of the microbial inoculates was undertaken at a concentration of 10^6^ CFU/mL. Chitosan, PVA, H_10_, H_11_, H_13_, H_31_, H_31_/AgNP composites, and *Vancomycin* and *Amphotericin B* were prepared in dimethyl sulfoxide, which proceeds as a negative control. Thereafter, in a 96-well plate, dilutions from 1000 to 0.1 μg/mL were carried out. Every microplate well included 40 µL of the growth medium brain heart infusion (BHI), 10 µL of inoculum, and 50 µL of the diluted sample. Afterwards, the plates of the tested microorganisms were incubated at 37 °C for 24 h. Then, 40 µL of tetrazolium salt (2,3-bis(2-methoxy-4-nitro-5-sulfophenyl)-2H-tetrazolium-5-carboxanilide) (XTT) were added to every well. Then, the plates were placed in the dark at 37 °C for 1 h for their incubation. A Tecan Sunrise absorbance microplate reader (UK) was utilized to measure the colorimetric alteration in the XTT reduction assay at 492 nm.

#### 2.2.6. Biofilm Inhibition Assay

Biofilms of the microbes were inhibited in 96-well plates in accordance with the assay for the eradication of the biofilms [35]. Mueller–Hinton broth (MHB) was used to incubate the microbial strains at 37 °C for 20 h, and then the cultures were diluted to a concentration of 1 × 10^7^ CFU/mL. The microbial cultures (150 µL) were placed in 96-pegs-lids, and their incubation for 42 h was performed for the permission of biofilm formation. The pegs were rinsed twice using the sterile phosphate-buffered saline (PBS) to remove the planktonic cells. The peg-lids were conveyed to another 96-well plate that included from 1000 to 0.98 μg/mL concentrations of the samples, and they were incubated at 37 °C for 4 h. PBS was used to rinse the pegs that were then conveyed to a new 96-well plate containing 3-(4,5-dimethyl-2-thiazolyl)-2,5-diphenyl-2H-tetrazolium bromide (MTT) in PBS (200 µL, 0.5 mg/mL) and they were incubated at 37 °C for 4 h. The aqueous solution of sodium dodecyl sulphate (50 µL, 25% *v*/*v*) was used for dissolving the crystals of formazan in every well. The sample’s capability for inhibiting the formation of biofilms formation was evaluated by measuring the reduced MTT to its insoluble formazan at 595 nm using a Microplate reader (TECAN, Inc., San Jose, CA, USA) to determine the viability of the biofilm cells. The minimum biofilm inhibitory concentration (MBIC) is the least concentration demanded to inhibit the biofilm after the sample’s treatment.

#### 2.2.7. Cytotoxicity Assessment Using Viability Assay

A 96-well plate was used to plant the normal human lung fibroblast cells (MRC-5 cells) of 1 × 10^4^ cells concentration per well in the growth medium (100 μL). After 24 h of planting, the test samples with various concentrations were added to freshly prepared media. Serial two-fold dilutions of the tested chemical compound were mixed with the confluent cell monolayers and then distributed, using a multichannel pipette, into a 96-well with flat-bottomed microtiter plates (Falcon, NJ, USA). A humidified incubator was used to incubate the microtiter plates with 5% carbon dioxide at 37 °C for 48 h. The control cells without test samples and with or without DMSO were incubated at 37 °C for 24 h. Afterwards, various sample concentrations were added and then continuous incubation for 24 h. Then, 1% crystal violet solution was added to the wells for 30 min and then the glacial acetic acid (30%). The absorbance of the plates was assigned at 490 nm after gentle shaking on a Microplate reader (TECAN, Inc., San Jose, CA, USA). The microplate reader (SunRise, TECAN, Inc., San Jose, CA, USA) was used to measure the optical density for an estimation of the number of viable cells. The viability percent can be calculated using Equation (2).
Viability (%) = [(ODt/ODc)] × 100%(2)
where ODt and ODc are the mean optical density of wells treated with the tested sample and the mean optical density of untreated cells, respectively.

The 50% inhibitory concentration (CC_50_) is defined as the concentration required for causing toxic impacts in 50% of the intact cells. It can be determined from the graphic plot of the curve of the dose-response for every concentration using GraphPad Prism software (San Diego, CA, USA) [36].

### 2.3. Measurements

#### 2.3.1. Elemental Analyses

Elemental analyses of chitosan, PVA, H_10_, H_11_, H_13_, and H_31_ were performed on a Perkin Elmer C, H, N, S Analyzer, Model 2410 series II (Waltham, MA, USA). 

#### 2.3.2. FTIR Spectroscopy

FTIR spectroscopy measurements of chitosan, PVA, H_10_, H_11_, H_13_, and H_31_ and the AgNP bio-composites were performed on an Agilent Technologies FTIR Spectrometer, Cary 600 Series (Santa Clara, CA, USA) in the wavenumber range from 4000 to 400 cm^−1^ in 16 scans. 

#### 2.3.3. X-ray Photoelectron Spectroscopy (XPS) 

X-ray photoelectron spectroscopy (XPS) measurement of the prepared H_31_/AgNPs5% composite (as a representative example of silver nano bio-composites) was performed on a Kratos XSAM spectrometer (Model 800, Manchester, UK) with MgKα X-ray source (12.5 kev, 12 mA and 150 W) and under 5 × 10^−7^ Torr operating pressure. The composite was exposed to an X-ray source for a minimum of twenty runs. Collection and analysis of the XPS data were performed using a Tektronix 4105 computer.

#### 2.3.4. X-ray Diffractometry

The advanced wide-angle X-ray diffractometer (Brucker’s D-8), having a nickel monochromator, was utilized to identify the inner structure of chitosan, PVA, H_10_, H_11_, H_13_, and H_31_ and the AgNP bio-composites at room temperature. CuKα radiation of 40 kV and 30 mA is the X-ray source. The samples were powder of constant weight and scanned in the reflection mode with a scattering angle 2θ ranging between 3° and 90° at a scanning rate of 8°/min. 

#### 2.3.5. Scanning Electron Microscopy (SEM)

The surface morphologies of chitosan, PVA, H_10_, H_11_, H_13_, and H_31_ were investigated using a scanning electron microscope (Jeol-JSM-6060LV) after their coating with gold and before photographing. 

#### 2.3.6. Energy-Dispersive X-ray Spectroscopy (EDS)

The AgNPs that formed in-situ inside the matrix of H_31_ (H_31_/AgNPs3%, as a representative example of silver nano bio-composites) were detected using a scanning electron microscope (Quanta FEG 250) equipped with an energy dispersive X-ray spectrometer (EDS).

#### 2.3.7. Transmission Electron Microscopy (TEM)

The transmission electron microscopy (TEM) measurements were employed for the characterization of the structural morphology of H_31_/AgNP composites. Their dispersed solutions were observed using a transmission electron microscope (JEM-HR-JEOL-JEM 2100) that operating at 120 kV.

## 3. Results and Discussion

### 3.1. Synthesis of Chitosan/PVA Hydrogels and H_31_/AgNP Composites

A series of novel hydrogels based on blends of different proportions of chitosan and PVA (H_13_, H_11_, and H_31_, Table 1) was synthesized via a chemical crosslinking using trimellitic anhydride isothiocyanate in addition to the possible H-bonds formed between chitosan and PVA. The crosslinking reaction in the blended polymers occurred between the amino groups of chitosan and both the anhydride ring and the isothiocyanate group of the crosslinker molecule. Crosslinking of pure chitosan with trimellitic anhydride isothiocyanate, which produced H_10_, was confirmed by its non-solubility in acidic media in addition to the appreciable changes in its elemental analysis (Table 1) and FTIR spectrum (Figure 1). In comparison, PVA could not crosslink with trimellitic anhydride isothiocyanate due to the poor activity of its hydroxyl groups relative to the amino groups of chitosan. This was proven from similar elemental analyses and FTIR spectra of PVA before and after its treatment with trimellitic anhydride isothiocyanate.

H_31_ was additionally modified via its impregnating with silver nanoparticles that were generated in situ in its matrices. The green preparation of silver nanoparticles in situ inside H_31_ was performed via both the functional groups of crosslinked chitosan and hydroxyl groups of PVA that enhanced the reducing activity of H_31_ (Figure 3).

### 3.2. Characterization of Chitosan/PVA Hydrogels and H_31_/AgNP Composites

#### 3.2.1. Elemental Analyses

Table 1 shows the elemental analyses of chitosan, PVA, H_10_, H_13_, H_11_, and H_31_. It was found that both the percentage of nitrogen and sulphur increased with the increasing crosslinked chitosan (H_10_) content in the hydrogels, i.e., from H_13_ to H_31_. On the other hand, both the percentage of carbon, hydrogen, and oxygen increased with the increasing PVA content in the hydrogels, i.e., from H_31_ to H_13_. This indicates the successful formation of the chitosan/PVA hydrogels.

#### 3.2.2. FTIR Spectroscopy

FTIR spectroscopy of chitosan (Figure 1) confirmed the existence of four absorption peaks corresponding to its saccharide moieties at 1155, 1073, 1030, and 889 cm^−1^. There is a broad, strong absorption peak ranging from 3600–3000 cm^−1^ attributed to -OH, -NH, inter- and intra-chains hydrogen bonds. In this part, the primary amino groups showed two absorption peaks at 3352 and 3201 cm^−1^. The peaks at 2915 and 2855 cm^−1^ are assigned to the stretching vibration of C-H. Further, the absorption peaks (weak) that concerned amide I and amide II appeared at 1650 and 1566 cm^−1^, respectively, indicating that the deacetylation degree of chitosan is high. The deforming absorption peak of the amino groups at 1600 cm^−1^ has overlapped with that of the amide I at 1650 cm^−1^, attaining a strong one [11].

FTIR spectrum of PVA (Figure 1) showed a broad peak at 3428 cm^−1^ attributed to stretching vibration of O-H groups, including both their inter- and intrachain hydrogen bonds. The stretching vibration of C-H groups appeared at 2929, 2847, and 1648 cm^−1^ [37], while the peak of CH_2_ groups appeared at 1438 cm^−1^. The stretching vibration of C-OH appeared at 1086 cm^−1^ [37]. 

After the blending of chitosan and PVA followed by the cross-linking of chitosan with trimellitic anhydride isothiocyanate cross-linker, the doublet peak of the primary amine groups at 3352 and 3201 cm^−1^ had disappeared, and was completely replaced by a single peak at 3291cm^−1^ (H_10_), 3296 cm^−1^ (H_31_), 3288 cm^−1^ (H_11_), and 3260 cm^−1^ (H_13_) related to the NH groups, confirming the reaction of all the amino groups of chitosan with the cross-linker (Figure 1). A new absorption peak was observed at 1717 cm^−1^ (H_10_), 1723 cm^−1^ (H_31_), 1715 cm^−1^ (H_11_), and 1710 cm^−1^ (H_13_), corresponding to COOH of the cross-linking linkages. In contrast, the absorption peak of the OH groups and COOH of the cross-linking linkages overlapped with that of the NH groups (Figure 1). The overlapping between the absorption peaks of the -CONH, NH (secondary amide), and aromatic C=C bonds resulted in an appearance of a new peak at 1655 cm^−1^ (H_10_), 1656 cm^−1^ (H_31_), 1656 cm^−1^ (H_11_), and at 1657 cm^−1^ (H_13_). The absorption peak of the C=S groups that overlapped with the stretching vibration of C-O appeared as a strong, broad peak at 1055 cm^−1^ (H_10_), 1086 cm^−1^ (H_31_), 1086 cm^−1^ (H_11_), and 1085 cm^−1^ (H_13_). The bending vibration peaks of N-C-S groups were observed at 1424 and 588 cm^−1^ (H_10_), at 1423 and 592 cm^−1^ (H_31_), at 1423 and 592 cm^−1^ (H_11_), and at 1424 and 588 cm^−1^ (H_13_) (Figure 1). The intensity of the peaks correlated to the cross-linker increased with the increasing cross-linker linkages content and consequently with increasing the chitosan content in the prepared hydrogels, i.e., from H_13_ to H_31_. The cross-linking process of chitosan is illustrated in Figure 2. 

Furthermore, FTIR spectra of silver nano bio-composites, H_31_/AgNPs3% and H_31_/AgNPs5% showed additional peaks in the range of 500–800 cm^−1^ [4], confirming the formation of silver nanoparticles into the matrices of the bio-composites (Figure 2).

#### 3.2.3. X-ray Photoelectron Spectroscopy (XPS) 

Further evidence for proving AgNPs formation into the matrices of H_31_ came from the recording of the XPS spectrum of H_31_/AgNPs5% composite, as illustrated in Figure 3. In this figure, there are four peaks at 368.2, 370.3, 374.0, and 376.1 eV. The first and the third peaks are in good agreement with 3d_5/2_ and 3d_3/2_ for the pure Ag metal as listed in the XPS spectrometer manual (at 367.9 and 373.9 eV, respectively). Whilst the second and the fourth peaks are due to the complex structure of the interacted silver with the oxygen of the carbonyl of the carboxylic groups and also with the hydroxyl groups in H_31_.

#### 3.2.4. X-ray Diffractometry (XRD)

The morphological structure of chitosan, H_10_, PVA, H_13_, H_11_, and H_31_ were inspected using an X-ray diffractometer. Figure 4 illustrates the changes that happened in the matrix of chitosan after blending with PVA and cross-linking. The virgin chitosan is distinguished by an inner structure involving amorphous and crystalline regions, which manifested in its XRD pattern as two peaks near 2θ = 10° and 20°, respectively [38,39]. This is ascribed to its capability to form a considerable amount of inter- and intra-chain hydrogen bonding due to its possession of many hydroxyl and primary amine polar groups. The modification of chitosan with trimellitic anhydride isothiocyanate cross-linker (Figure 1) leads to a remarkable reduction in the amount of hydrogen bonding due to the consumption of its amino groups during the cross-linking process. This led to a separation of the chitosan chains away from each other, an increment in the amorphous region, and a reduction in the crystalline fraction. This is shown from the evanescence of the peak at 2θ = 10° and a broadening and a notable lowering of the intensity of the peak at 2θ = 20° in the XRD pattern of H_10_ (Figure 4). XRD pattern of virgin PVA showed three peaks; the first peak is very weak and broad that appeared near 2θ = 10°, the second peak is very intensive and sharp at 2θ = 20°, and the third peak is weak and broad that manifested at 2θ = 40° [40]. On the other hand, XRD patterns of H_13_, H_11_, and H_31_ included three relatively amorphous peaks at 2θ = 10°, 20°, and 40°. Their intensities and sharpness increased with increasing the PVA content in the blends from H_31_ to H_13_ (Figure 4).

In order to confirm the formation of AgNPs into the matrices of H_31_, XRD measurements were performed for two of the prepared Ag nano bio-composites (Figure 5). All the patterns of H_31_/AgNP composites showed five new peaks, in addition to the amorphous peak of H_31_ near 2θ = 20°, at 2θ = 38.18°, 44.25°, 64.55°, 77.58°, and 81.81° with plane distances of 2.357°A, 2.039°A, 1.444°A, 1.229°A, and 1.165°A, respectively. These peaks and their corresponding plane distances are in good agreement with those for the pure silver metal (2θ = 38.14°, 44.33°, 64.50°, 77.61°, and 81.85° with plane distances = 2.359°A, 2.043°A, 1.445°A, 1.230°A, and 1.166°A, respectively) as reported by ASTM [41]. This evidences the formation of AgNPs within the matrices of H_31_. Figure 5 also demonstrates that the intensity of all the Ag metal peaks increased as a function of AgNPs content in H_31_. Further, the diffraction peaks were indexed to crystal planes of (111), (200), (220), (311), and (222) of face-centered cubic (fcc) Ag, which were consistent with that of Ag (JCPDS No. 4-0783). From the crystallinity data, the crystal structure of AgNPs was fcc.

#### 3.2.5. Scanning Electron Microscopy (SEM)

Inspection of the morphology and topography of H_13_, H_11_, and H_31_ was conducted utilizing the scanning electron microscopy (SEM) technique, and the SEM images are displayed in Figure 6. The SEM images of the parent chitosan, H_10_, and virgin PVA are also given for comparison. While both the chitosan and PVA have lump-free smooth surfaces, all the H_10_, H_13_, H_11_, and H_31_ possess rough surfaces, full of grooves and containing many pores resulting from crosslinking networks. It can be noted that the grooves were homogeneously distributed, referring to the successful completion of the crosslinking process. These crosslinks cause a dropping in the inter-chain hydrogen bonds and lead to a separation of the polymeric chains from each other, obtaining a much more open structure with increased surface area. It is postulated that these pores facilitate the penetration of water into the hydrogel and act as sites of interaction of the external stimuli with the hydrophilic polar groups of the hydrogels. Although the surfaces of all the prepared hydrogels have a similar appearance, the size and distribution of the pores varied from H_13_-H_31_. The pores were denser and more regular as the chitosan content in the hydrogel increased due to the increased concentration of the used crosslinker (Table 1). This allows the formation of a microporous structure having a large surface area and a great swell ability.

In addition to the color change of the hydrogel upon the formation of AgNPs, the morphological structure of the synthesized H_31_/AgNP composites was also inspected using SEM. It can be noted that the in situ created AgNPs were homogeneously distributed as bright spots into the observed composites (Figure 7). They dispersed without agglomeration or aggregation into the polymeric matrices. This is due to the presence of many marked capping groups on the H_31_, which work as efficient ligands for the stabilization of the generated AgNPs. Further, the AgNPs electrostatically interacted with the functional groups of H_31_, resulting in an adequately homogeneous distribution of the produced AgNPs. 

#### 3.2.6. Energy-Dispersive Spectroscopy (EDS)

To assert the in situ forming AgNPs into H_31_, an EDS measurement was also performed, as illustrated in Figure 7. In the EDS diagram for H_31_/AgNPs3% (as a representative example), the peak corresponding to silver is attributed to the dispersed AgNPs. The quantity of silver was 5.98 wt%, indicating the considerable adsorption of AgNPs into H_31_. 

#### 3.2.7. Transmission Electron Microscopy (TEM)

The morphology of the nanocomposites was also inspected utilizing the TEM technique to emphasize the formation of AgNPs inside H_31_ and assist in describing the characteristics of the structure of the AgNPs. TEM images of H_31_/AgNPs5% composite at different magnifications were shown in Figure 8a,b, which clarified the homogeneous dispersion of the AgNPs as spherically shaped spots. Their sizes ranged between 13 and 26 nm in the matrices of H_31_ (Figure 8). Furthermore, there is no agglomeration or aggregation of AgNPs into the matrices of the H_31_ because of the interaction between AgNPs with the functional groups of the H_31_, indicating the role played by the H_31_ as a stabilizer of the AgNPs.

#### 3.2.8. Solubility Behavior

The solubility of chitosan, PVA, H_10_, H_13_, H_11_, H_31_, and H_31_/AgNP composites was checked at 60 °C in a variety of solvents such as water, 1% aqueous solution of acetic acid, N,N-dimethyl formamide, tetrahydrofuran, chloroform, ethanol, acetone, and methylene chloride. The results revealed that H_10_, H_13_, H_11_, H_31_, and H_31_/AgNP composites did not dissolve in any of the used solvents since their initial weights were completely retained without any loss. On the other hand, the virgin chitosan dissolved only in the 1% aqueous solution of acetic acid and did not dissolve in water or the other utilized solvents [42]. This is ascribed to its basic nature since it possesses numerous primary amine groups along its chains. In comparison, PVA dissolved only in water and did not dissolve in the other used solvents. This solubility conduct emphasized the successful preparation of the hydrogels. 

### 3.3. Antimicrobial Activity

Tetrazolium salt, having a negative charge (XTT), is reduced into the orange, soluble formazan dye. The reduction process takes place outside the cell by the transportation of the electrons through the membrane of the plasma of the living cell. The quantity of XTT indicates the metabolic activity of the cell. For evaluating whether the metabolic activity of the cell has reduced or increased, its absorbance should be measured and compared with that of the control solution of the non-treated cell. Thus, XTT is utilized to evaluate cell proliferation or to determine drug cytotoxicity.

An XTT test was carried out to screen the activity of the chitosan, PVA, H_10_, H_13_, H_11_, H_31_, and H_31_/AgNP composites against different microbial strains by determining the values of the minimum inhibitory concentration (MIC), which inhibits microbe growth.

#### 3.3.1. Antibacterial Activity

The antibacterial activities of chitosan, PVA, H_10_, H_13_, H_11_, H_31_, and H_31_/AgNP composites were studied against some Gram-negative bacteria, namely: *Klebsiella pneumonia* (*K. pneumonia*, ATCC 13883), *Proteus mirabilis* (*P. mirabilis*, ATCC 12453), *Escherichia coli* (*E. coli*, ATCC 11775), *Pseudomonas aeruginosa* (*P. aeruginosa*, ATCC 10145), and *Acinetobacter baumannii* (*A. baumannii,* ATCC 19606), and against some Gram-positive bacteria, namely: *Bacillus subtilis* (*B. subtilis*, ATCC 6051), *Staphylococcus epidermidis* (*S. epidermidis*, ATCC 11774), *Staphylococcus aureus* (*S. aureus*, ATCC 25923), *Methicillin-resistant Staphylococcus aureus* (*MRSA*, ATCC- BAA-1720), and *Streptococcus pyogenes* (*S. pyogenes*, ATCC 12344) using an XTT inspection. For a comparison, *Vancomycin*, as an example of the traditional antibacterial drugs, was also studied.

The obtained results exhibited that PVA has no Inhibition activity against any of the tested microorganisms. This may be attributed to its weak acidic nature due to its hydroxyl groups. Thus, PVA cannot interact with the negatively charged membranes of bacteria and fungi [6,7]. On the contrary, H_10_ showed a much higher antibacterial activity than that of virgin chitosan (Figure 9a and Figure 10a).

Three mechanisms that describe the action of chitosan against bacteria have already been suggested. The most sensible one indicates that the polycationic chitosan chains electrostatically interact with polyanionic bacterial cell membranes. This leads to a change in the permeability of these membranes and a loss in the internal electrolytes and the proteineous ingredients of the bacterial cells [43]. In accordance with this proposition, the antibacterial efficacy of the materials will improve as their cationized sites increase. In this respect, the -COOH, -CONH- and -CO-NH-CS-NH- linkages that have been incorporated into chitosan via the cross-linking process, producing H_10_, results in an increment of its cationic sites. This is due to the easy protonation of their carbonyl, hydroxyl, amino, and thiocarbonyl groups, increasing the total positive charge intensity, raising their electrostatic interaction with bacterial cell membranes, which have anionic sites, and enhancing the activity of H_10_ against bacteria compared to the parent chitosan. This conclusion is in good agreement with that have previously been published in the literature [30]. 

In the second mechanism, it was assumed that chitosan is amalgamated with the DNA of the bacteria, thus, restraining the synthesis of the protein and mRNA [44]. In H_10_, the great separation of the polymeric chains from each other, due to the inclusion of the long, polar cross-linking linkages, reduces its inter-chains hydrogen bonds and increases its hydrophilicity and swelling capacity. This enhances the penetration of H_10_ into the bacterial cells and suppresses their growth by inhibiting the transformation of DNA to RNA, realizing a greater antibacterial efficacy compared to virgin chitosan.

The third mechanism was proposed dependently on the characteristic ligand ability of chitosan with metal salts, major nutritious materials, and spore elements [45]. It was earlier reported that the -COOH, -CONH-, and -CO-NH-CS-NH- groups possess a high capability for metals chelation [46]. Thus, the incorporation of these groups into H_10_ increases its chelating sites for metal salts, major nutritious materials, and spore elements. This explains the high potency of H_10_ against bacteria compared with the parent chitosan.

All the prepared hydrogels, H_13_, H_11_, and H_31_, showed a considerable inhibition potency against all the tested Gram-negative (Figure 9a) and Gram-positive bacteria (Figure 10a), which increased with the increasing H_10_ content in the hydrogels; from H_13_ to H_31_. 

Furthermore, the inhibitory performance for the prepared hydrogels against the examined Gram-negative bacteria could be assorted as follows: *A. baumannii* > *P. aeruginosa* > *E. coli* > *P. mirabilis* > *K. pneumonia* (Figure 9a). While their performance in inhibiting the Gram-positive bacteria activity could be arranged as follows: *B. subtilis* > *MRSA* > *S. epidermidis* > *S. aureus* > *S. pyogenes* (Figure 10a). 

In general, the prepared hydrogels have a better inhibition potency against the tested Gram-positive bacteria than that against the Gram-negative bacteria. This may be ascribed to the structural differences in the cell walls of the diversified bacterial strains. The walls of the Gram-positive bacterial cells are highly porous, allowing diffusion of the foreign matters easily into the cells and their adsorption rapidly. Whereas the structures of the walls of the Gram-negative bacterial cells are complex and made of two layers of membrane; a thin interior membrane and a thick exterior membrane. The latter acts as a barrier to prevent foreign matter from diffusing into the cell [44]. Thus, the variance in the structures of the walls of these two sorts of bacterial cells is responsible for the different effectiveness exhibited by the hydrogels. Thus, the incorporation of PVA with crosslinked chitosan in the same hydrogel significantly improves PVA antibacterial activity, which is strongly influenced by the crosslinked chitosan content in the hydrogel. As a result, there is a great possibility to prepare PVA hydrogels that have considerably different antibacterial activities by adjusting the crosslinked chitosan content in these hydrogels.

Although H_31_ is the most potent hydrogel for inhibiting the activity of the tested microbes in comparison to the other two hydrogels, H_11_ and H_13_, its inhibition efficiency is lower than that of the commonly used drugs, *Vancomycin and Amphotericin B*. So, H_31_ has been selected for the preparation of AgNP composites to enhance its antimicrobial activity.

As would be expected, the inclusion of AgNPs inside the matrix of H_31_ remarkably enhanced its potency against the activity of all the examined bacterial species, as illustrated from the MIC values. It can be seen that H_31_/AgNPs1%, H_31_/AgNPs3%, and H_31_/AgNPs5% displayed MIC values that range from 21.25 to 0.10 µg/mL against all the examined Gram-negative (Figure 9b) and Gram-positive bacteria (Figure 10b), which are much lower than the MIC values of 62.50 to 2.38 µg/mL exhibited by their parent H_31_.

Further, the inhibitory action of H_31_/AgNPs3% against all the examined bacteria is better than that of H_31_/AgNPs1%. The MIC values of H_31_/AgNPs3% were 10.00, 6.88, 3.75, 2.25, and 1.87 µg/mL against *K. pneumonia*, *P. mirabilis*, *E. coli*, *P. aeruginosa*, and *A. baumannii*, respectively, which are lower than the MIC values of 21.25, 15.63, 11.25, 7.50, and 3.75 µg/mL for H_31_/AgNPs1%. Moreover, H_31_/AgNPs3% showed MIC values equal to 2.38, 1.50, 1.00, 0.63, and 0.13 µg/mL against *S. pyogenes*, *S. aureus*, *S. epidermidis*, *MRSA*, and *B. subtilis*, respectively, which were lower than the MIC values of 7.50, 2.13, 1.63, 1.25, and 0.38 µg/mL for H_31_/AgNPs1%. No perceivable enhancement in the inhibition potency of the AgNPs bio-composites was observed by increasing the inserted AgNPs from 3% to 5%. In comparison with the standard drug *Vancomycin*, H_31_/AgNPs3% showed almost analogous activity against *E. coli* and a higher activity against *P. aeruginosa*, *A. baumannii*, and all the tested Gram-positive bacteria (Figure 9b and Figure 10b).

There are several mechanisms that have been proposed to explain AgNPs’ action against bacteria. The most reasonable one is based on the adhesion of AgNPs to the membranes of bacteria, penetrating inside the cells of bacteria, generating free radicals, and modifying the pathway of microbial signal transduction [47]. Moreover, AgNPs can prevent microbial cell division, leading to deterioration in the envelope of bacterial cellular content [48]. In addition to the capability of AgNPs to induce hydroxyl radicals which are used in the killing of bacterial cells. Nano-composites comprising AgNPs display considerable activity against microbes, primarily by releasing silver ions and secondly via the production of reactive oxygen species. This leads to instantaneous damage to the cell membranes of microbes, resulting in an increment of the permeabilities of these membranes, a loss in proton motivation force, cell de-energization, a flowing out of phosphate, contents of the cell leaking out, disruption to ATP production, and disturbance to DNA replication [49,50]. Consequently, the activity of the AgNPs bio-composites against bacteria is attributed to a synergistic impact that takes place between the antibacterial activity of both the H_31_ and AgNPs in the composites. Thus, the investigated AgNPs bio-composites could be a prospective alternative for antibiotics, particularly against the bacterial strains having resistance to traditional drugs.

#### 3.3.2. Antifungal Activity

The conduct of chitosan, H_10_, PVA, H_13_, H_11_, and H_31_ and H_31_/AgNP composites was studied against *Aspergillus fumigatus* (*A. fumigatus*, ATCC 9197), *Syncephalastrum racemosum* (*S. racemosum* ATCC 14833), *Aspergillus niger* (*A. niger*, ATCC 6275), *Cryptococcus neoformans* (*C. neoformans*, ATCC 66031), and *Candida albicans* (*C. albicans*, ATCC 18804) as representative examples for fungi using an XTT assay. *Amphotericin B*, as a traditional antifungal drug, was utilized for comparison.

The results revealed that no inhibition activity of PVA against the examined fungi was observed. On the other hand, H_10_ exhibited much greater antifungal activity than the parent chitosan, as shown in Figure 11a. This is demonstrated via its decreased MIC values acquired from the calorimetrically change in the XTT color. In comparison to chitosan, H_10_ showed extremely decreased MIC values of 16.25, 11.25, 6.25, 2.38, and 1.88 μg/mL against *S. racemosum*, *C. neoformans*, *A. niger*, *A. fumigatus*, and *C. albicans*, respectively. It is well known that chitosan can inhibit spores from germination, retard the germ tube from elongation, and hinder the germ radial from growth [51]. The mechanism of the activity of chitosan against fungi includes a modification of the morphology of the cellular wall by chitosan, leading to a direct intervention in the fungi growth, similar to the impacts recognized in the cells of bacteria [51]. The microscopic observation confirmed that chitosan disseminates inside the fungal hypha and interferes with enzymes that are accountable for the growth of fungi [52]. The appropriate mechanism that interprets the greater antifungal activity of H_10_ as compared with parent chitosan may be due to: (1) The incorporation of the -COOH, -CONH-, and -CO-NH-CS-NH- linkages between chitosan chains, during the crosslinking process, decreases its crystallinity and increases its hydrophilicity and swelling ability. This facilitates its penetration and diffusion into the hyphae of fungi, disrupting the activities of the enzymes that are accountable for standard growth and leading to the absorption of insoluble materials on the surface of the hyphae of the fungi. (2) The -COOH, -CONH-, and -CO-NH-CS-NH- linkages possess a higher capacity for chelation with the metal ions necessary for nutrition and growth of the fungi.

Interestingly, the synthesized hydrogels, H_13_, H_11_, and H_31_, displayed a substantial inhibition performance against all the tested fungi, which increased with the increasing H_10_ proportion in the hydrogels; from H_13_ to H_31_ (Figure 11a). In addition, the order of the inhibition action of these hydrogels against the tested fungi was as follows: *C. albicans* > *A. fumigatus* > *A. niger* > *C. neoformans* > *S. racemosum*. This indicates that the cell walls of the five tested fungi are structurally different, leading to a diverse impact of the hydrogels on these fungi (Figure 11a). Again, the amalgamation between PVA and crosslinked chitosan in the same hydrogel matrix considerably boosts the PVA antifungal efficacy that depends on the content of the crosslinked chitosan in the hydrogel. Thus, it is possible to synthesize PVA hydrogels of high antifungal performance via an adjustment of the cross-linked chitosan content in these hydrogels.

The insertion of AgNPs into the matrix of H_31_ considerably improved its efficiency in inhibiting the activity of all the examined fungal strains, as illustrated by the MIC values (Figure 11b). H_31_/AgNPs1%, H_31_/AgNPs3%, and H_31_/AgNPs5% showed MIC values that ranged from 13.75 to 0.72 µg/mL against all the examined fungi, which are lower than the MIC values of 22.50 to 2.50 µg/mL that were obtained by their parent H_31_. Thus, it is assumed that AgNPs diffuse inside the fungal hyphae, destroy the plasma membrane, and inhibit enzyme activity that is accountable for the growth of fungi [47,50].

Again, against all the examined fungi, H_31_/AgNPs3% was more potent than H_31_/AgNPs1%. The MIC values of H_31_/AgNPs3% were 6.25, 2.13, 1.50, 1.00, and 0.75 µg/mL against *S. racemosum*, C. *neoformans*, *A. niger*, *A. fumigatus*, and *C. albicans*, respectively, which were lower than the MIC values of 13.75, 6.25, 2.25, 1.75, and 1.50 µg/mL for H_31_/AgNPs1%. There was no appreciable improvement in the inhibitory performance of the AgNPs bio-composites when the inserted AgNPs increased from 3% to 5%. Compared with the standard drug *Amphotericin B*, H_31_/AgNPs3% showed higher activity against all the tested fungi (Figure 11b). Thus, the inhibitory action of the AgNPs bio-composites is due to the synergistic impact that takes place between the antifungal activity of both H_31_ and AgNPs in the composites. This makes these AgNP bio-composites a prospective alternative for antibiotics, particularly against fungal strains that have a resistance to traditional drugs.

### 3.4. Biofilm Inhibition

Now it has widely been established that the growth pattern of the biofilms is the predominating pattern for the growth of bacteria either in the host or the environment. Thus, the antimicrobial agents should be characterized by inhibiting not only the growth of the cells of bacteria but also the growth of the biofilm.

The results revealed that no inhibitory action of PVA against biofilm formation of the examined microbes was observed. The results of the inhibitory performance of chitosan, H_10_, H_13_, H_11_, H_31_, H_31_/AgNPs1%, and H_31_/AgNPs3% against forming biofilms by *A. baumannii*, *B. subtilis*, and *C. albicans* are shown in Table 2 and Figure 12. 

The results revealed that chitosan has a poor inhibitory action against the formation of biofilms when compared with that of H_10_. The minimum biofilm inhibitory concentration (MBIC) values of chitosan (ranging from 500 to 1000 µg/mL) are extremely larger than those of H_10_ (ranging from 7.81 to 15.63 µg/mL) (Table 2). In contrast, H_13_, H_11_, and H_31_ exhibited a distinct capability for inhibiting the formation of biofilms, which increased by increasing the crosslinked chitosan content in the hydrogel; from H_13_ to H_31_. Their MBIC values ranged from 15.63 to 125 µg/mL (Table 2).

The dispersion of AgNPs inside the matrix of H_31_ appreciably enhanced its capacity for inhibiting the formation of biofilms, as illustrated in the MBIC values (Table 2). H_31_/AgNPs1% and H_31_/AgNPs3% showed MBIC values that ranged from 1.95 to 7.81 µg/mL, which were lower than the MBIC values of 15.63 to 31.25 µg/mL that were obtained by their parent H_31_. Moreover, H_31_AgNPs3% was more potent than H_31_/AgNPs1% in inhibiting biofilm formation (Table 2). 

Quorum sensing (QS) is a mechanism of communication between microorganisms that permits particular controlled processes for the formation of biofilms. It depends on a constant secretion of signalling molecules into the environment. The prime function of QS is the regulation of vital cell processes, such as the formation of biofilms or the production of a factor of virulence [53,54]. Due to the increment in the resistance of microbes to antibiotics, particularly those which were overused, there is a serious necessity to discover substitutional antimicrobial therapies; among them is the quorum quencher (QQ), which obstructs the communication of microbes [55].

On the basis of this approach, the polycationic chitosan has a capacity for interaction with the negatively charged membranes of the cells of microbes, disrupting their functions and thus inhibiting the communication of the microbes. This leads to a lowering in the signalling molecules secretion that delays the formation of biofilms and production of a factor of virulence. On the other hand, the H_10_ and consequently H_13_, H_11_, H_31,_ and AgNPs bio-composites were more potent in inhibiting the activity of the tested microbes than chitosan (Figure 9, Figure 10 and Figure 11) because they are more polycationic than chitosan due to they possess numerous additional functional groups (-COOH, CONH, NH-CS-NH-CO-). Thus, these hydrogels would easily penetrate and intervene in the QS signals better than chitosan. So, they are deemed to be more efficient in QS disturbance and accordingly effectively prevent the formation of biofilms. This conclusion is supported by some recent studies which depict the capability of chitosan to reduce the regulation of the expression of the regulated genes of QS in *P. aeruginosa* [56,57]. 

### 3.5. Cytotoxicity Evaluation

Figure 13 showed the results of the cytotoxicity inhibitory effect of H_31_ and H_31_/AgNPs3% against normal human lung fibroblast cells (MRC-5 cells). The samples were tested at a similar range of concentrations from 0 to 1000 µg/mL. The cells viability is not influenced by both H_31_ and H_31_/AgNPs3% at a concentration less than 250 µg/mL and 15.6 µg/mL, respectively; in other words, there is no inhibition for both samples at less than these concentrations. The half-calculated concentration of cytotoxicity (CC_50_), the concentration in demand for causing toxic impacts in 50% of intact cells, for H_31_ and H_31_/AgNPs3% was 810.70 ± 14.50 and 62.50 ± 1.71 µg/mL, respectively. From Figure 9b, Figure 10b and Figure 11b, H_31_ and H_31_/AgNPs3% are characterized by high antimicrobial activities. They showed MIC values less than the concentrations of cytotoxicity of H_31_ (250 µg/mL) and H_31_/AgNPs3% (15.6 µg/mL), suggesting that H_31_ and H_31_/AgNPs3% are secure in normal human cells, and thus, they can be utilized in pharmaceutical applications as antimicrobial agents.

## 4. Conclusions

PVA was modified by blending it with chitosan in various weight ratios, followed by crosslinking using trimellitic anhydride isothiocyanate at a concentration based on the weight of chitosan in the blend. The weight ratios of chitosan: PVA were 1:3, 1:1, and 3:1 to obtain three novel hydrogels designated as H_13_, H_11_, and H_31_, respectively. For a comparison, H_10_ was prepared by crosslinking pure chitosan with trimellitic anhydride isothiocyanate. Three silver nano bio-composites denoted as H_31_/AgNPs1%, H_31_/AgNPs3%, and H_31_/AgNPs5% were also prepared. In comparison to PVA, which has no antimicrobial and antibiofilm formation activity, the investigated samples showed a distinguished inhibitory performance against all the tested microbes and against microbial biofilm formation, which can be arranged as H_31_/AgNPs5% ≥ H_31_/AgNPs3% > H_31_/AgNPs1% > H_10_ > H_31_ > H_11_ > H_13_ > chitosan. Further, they are more potent in inhibiting the activity of the Gram-positive bacteria activity than Gram-negative bacteria due to the structural difference in the cellular walls of these two types of bacteria. Interestingly, H_31_/AgNPs3% showed a better or even a comparable inhibition potency than that of the used standard drugs against the tested microbes. This is due to the synergism that occurs between the antimicrobial and antibiofilm activities of both H_31_ and AgNPs in the bio-composites. The results of the cytotoxicity evaluation revealed that H_31_ and H_31_/AgNPs3% are safe on normal human cells. Thus, one can conclude that the incorporation between PVA, crosslinked chitosan, and AgNPs in the same hydrogel matrix extremely reinforces the PVA efficacy against microbes and against microbial biofilm formation, depending on the content of both the crosslinked chitosan and AgNPs in the hydrogel. Thus, it is possible to synthesize PVA hydrogels of high antimicrobial and antibiofilm formation performance via adjustment of the crosslinked chitosan and AgNPs content in these hydrogels. This is a suitable approach to attain promising materials that can efficiently compete with conventional antibiotics.

## Data Availability

The data presented in this study are available on request from the corresponding authors.

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
