# Peer review of "Evaluation of Antimicrobial and Anti-Biofilm Formation Activities of Novel Poly(vinyl alcohol) Hydrogels Reinforced with Crosslinked Chitosan and Silver Nano-Particles"

_polymers, 2022, doi:10.3390/polym14081619_

Round 1

Reviewer 1 Report

The Authors prepared the manuscript well, however, many major concerns need to solve carefully before final publications. Please revise the manuscript accordingly.

The title needs to shorten and precise. In the title, the first character of each word should be capitalized as per the journal guidelines. Kindly check the previous paper and correct it.

Line 12-15 is difficult to understand for readers. please rewrite it.

Indicate clearly what are these, H13, H11, and H31 in” novel hydrogels symbolized as H13, H11, and H31".

In keywords, Synthesis; Characterization; these are too general. replace it with technical-scientific terms.

Materials: Line 123, what is code, Chitosan (code KB-002)?

Why do authors convert trimellitic anhydride chloride to trimellitic anhydride isothiocyanate? What are the advantages? Please discuss its importance. What is the mechanism of the reaction? how to confirm it.

Line 42 check doted mark.

Line 49, Pseudomonas aeruginosa and Staphylococcus aureus [2]. Microorganisms should be named should be written in italic. Check other please these types of errors.

Line 87-102 there are unnecessary references please remove it.

Line 100, check subscript in NH2.

Line 117, Please use the full name of FTIR… TEM then use short name in the whole manuscript.

In introduction section too many paragraphs make it short too many sentences. Avoid general things. Discuss other groups research work related to this study and highlighted the novelty.

Line 131-132 needs to rewrite correctly.

Line 143, should be space between 25 and oC "at 25°C overnight for complete". Check these types of errors throughout the manuscript.

Line 138, how to confirm byproducts was isolated by filtration.

Line 140 eq. 1. It is not an equation. It is the mechanism scheme.

After crosslinking the amino functional group decreases in chitosan. please calculate how much % decrease. Authors can follow and cite this paper to calculate it International journal of Biological macromolecules 136 (2019) 661-667.

Line 14 (5.0 %wt/v), please write correctly as 5 wt% v/v.

Line 145, how PVA dissolves in 60 oC? PVA melting point is higher than 60oC. Please correct it.

Line 161, write correctly as et al.

Line 154 to 155 please provide the optical images of hydrogen in the supplementary information.

Why authors choose only H31 sample to prepare H31 /silver nanocomposites? Is there any specific reason please indicate it inside the manuscript?

Line 162, write mol L-1.

Authors need to provide NMR data of cross-linked chitosan.

line 206, the sentence should not start with a number (150 µL) please rewrite it.

Line 222 104 please write clearly superscript.

In 2.3.2 in FTIR section, please write the reference and scanning number.

Line 254, -7 please write in the correct way. Check in whole manuscript these types of errors.

In all figures please clearly indicate major, minor ticks as well as x-axis and y- axis caption. All figures resolution is low.

Merge figures 1 and 2 into one. And all the curves place into a stack format. 

make all the bands and its assignment into a tabular format for clear understanding.

Line 306-313 need to insert citations. Please insert it this line or other places in manuscript. Carbohydrate Polymers, 2022, 287, 119318, Carbohydrate Polymers 257 (2021) 117633 Carbohydrate Polymers, 2021, 261, 117875 and Cellulose, 29 (2022) 2399-2411. Line 342. It is not an equation. It is a scheme. Please correct it accordingly other schemes.

Figure 3 is messy please present it clearly. Deconvulted spectra of Ag and S need to provide. Other samples XPS data also need to provide.

Line 485-493 can be rewritten in a better way.

Figure 4 is missing and merged into figure 5. Calculate all the sample's crystallinity data. Please follow and cite the following papers Journal of Luminoscense, 228 (2020) 117593.

figure 5 y-axis title is missing.

Figure 6 is missing. I am not sure why authors were not provided.

The authors have cited too many self-citations please reduce it.

Figure 7 please indicate the scale bar clearly.

In section 3.2.8, solubility behavior, what about others please provide the data if it swells or dissolves if dissolve calculates the exact amount. Additionally, when increasing the pH what is the effect of solubility in respective pH. please determine from UV data. Authors can follow this paper and cite Materials Letters 316 (2022) 132046.

Why PVA is not showing antibacterial and antifungal effects?

Why PVA/CS showing antibacterial and antifungal effects?

Table 3 need statistical analysis.

Table 3, footnote, line 718, ml should be written as mL. please correct other places in the manuscript.

Table 3 need statistical analysis. it is better to present it as a bar graph.

Merge Table 3 and 4 into one bar graph.

All figure charity so low. Author written figure but there are no figures 

The conclusions section needs to be shortened with important findings of this study.

Check references font style. It seems different from then the journal guidelines and the main text of the manuscript.

Author Response

Ms. Claire Zhang
Section Managing Editor
of Section "Polymer Physics and Theory"
Polymers Editorial Office

It gives me much pleasure to send you our manuscript (Manuscript ID: polymers-1666546) that has been revised according to all the reviewers’ comments. The changes were highlighted in yellow color.

With best regards

Reem T. Alfuraydi

Department of Chemistry

Faculty of Science

Qassim University

Top of Form

Comments and Suggestions for Authors,

The Authors prepared the manuscript well, however, many major concerns need to solve carefully before final publications. Please revise the manuscript accordingly.

(1) The title needs to shorten and precise. In the title, the first character of each word should be capitalized as per the journal guidelines. Kindly check the previous paper and correct it.

Our response: Done. (Please refer to the Title)

Evaluation of Antimicrobial and Anti-Biofilm Formation Activities of Novel Poly(Vinyl Alcohol) Hydrogels Reinforced with Crosslinked Chitosan and Silver Nano-Particles

(2) Line 12-15 is difficult to understand for readers. please rewrite it. Indicate clearly what are these, H13, H11, and H31in” novel hydrogels symbolized as H13, H11, and H31".

Our response: Done. Please refer to Line 10-13.  

Novel hydrogels were prepared by blending chitosan and poly(vinyl alcohol), PVA, then crosslinking the resulting blends using trimellitic anhydride isothiocyanate at a concentration based on chitosan content in the blends. The weight ratios of chitosan: PVA in the blends were 1:3, 1:1 and 3:1 to produce three hydrogels symbolized as H13, H11 and H31, respectively.  

(3) In keywords, Synthesis; Characterization; these are too general. replace it with technical-scientific terms.

Our response: Done. Please refer to Keywords.

Keywords: PVA/chitosan hydrogels; Trimellitic anhydride isothiocyanate crosslinker; Silver nanoparticles; Antimicrobial activity; Anti-biofilm formation activity; Cytotoxicity

(4) Materials: Line 123, what is code, Chitosan (code KB-002)?

Our response: It is the chitosan code. Now it has been omitted (Please refer to Line 136).

(5) Why do authors convert trimellitic anhydride chloride to trimellitic anhydride isothiocyanate? What are the advantages? Please discuss its importance. What is the mechanism of the reaction? how to confirm it.

Our response: Done (Please refer to line 113-119).

In our previous work, we have crosslinked parent chitosan using trimellitic anhydride isothiocyanate through reaction of both the anhydride and isothiocyante groups with the amino groups of chitosan, resulting in incorporation the bioactive thiourea and amide linkages in addition to carboxylic groups into chitosan. This considerably enhanced both the antimicrobial activity and the adsorption capacity for anionic Congo red and cationic Basic red 12 dyes of the produced cross-linked chitosan hydrogels [30-32].

The mechanism of the reaction of the anhydride and isothiocyante groups with the amino groups is well known, so there is no need to explain it.

In this work, novel PVA-based hydrogels incorporating chitosan crosslinked with trimellitic anhydride isothiocyanate were not previously prepared. The occurrence of the crosslinking reaction was confirmed using solubility test (the produced hydrogels were insoluble, section 3.2.8.), elemental analysis (the presence of Sulphur in the hydrogels as shown in Table 1), FTIR spectra (section 3.2.2.), X-ray diffraction (section 3.2.4.) and scanning electron microscopy (section 3.2.5.).

(6) Line 42 check doted mark.

Our response: The doted mark has been omitted (Please refer to Line 41)

(7) Line 49, Pseudomonas aeruginosa and Staphylococcus aureus [2]. Microorganisms should be named should be written in italic. Check other please these types of errors.

Our response: Done. (Please refer to Line 48-49).

(8) Line 87-102 there are unnecessary references please remove it.

Our response: Done (References [20], [21], [22], [24], [25], [26] have already been replaced by others suggested by the reviewers). Please refer to Line 90-103.

(9) Line 100, check subscript in NH2.

Our response: Done (Please refer to Line 110).

(10) Line 117, Please use the full name of FTIR… TEM then use short name in the whole manuscript.

Our response: Done. Please refer to Line 127-130.

Fourier-transform infrared spectroscopy (FTIR), X-ray photoelectron spectroscopy (XPS), X-ray diffraction (XRD), Energy-dispersive X-ray spectroscopy (EDS), Scanning electron microscopy (SEM) and Transmission electron microscopy (TEM).

(11) In introduction section too many paragraphs make it short too many sentences. Avoid general things. Discuss other groups research work related to this study and highlighted the novelty.

Our response: Done. Please refer to line 92-101concerning the discussion of other groups research work related to this study. The novelty of the present work was indicated in Line 119-120

(12) Line 131-132 needs to rewrite correctly.

Our response: Done. Please refer to Line 143-144.

Trimellitic anhydride chloride, the other used chemicals and solvents were extra pure and received from Aldrich (Germany).

(13) Line 143, should be space between 25 and °C "at 25°C overnight for complete". Check these types of errors throughout the manuscript.

Our response: Done.

(14) Line 138, how to confirm by products was isolated by filtration.

Our response: The by-product (a white precipitate, ammonium chloride) was confirmed by some simple tests:

(1) When the aqueous solution of the by-product was boiled in NaOH solution, ammonia gas evolved that formed white clouds when exposed to a glass rod wetted with HCl.

(2) When sodium cobaltinitrite solution was added to the aqueous solution of the by-product, a yellow precipitate was formed.

(3) When the aqueous solution of the by-product was boiled in concentrated sulfuric acid, HCl gas was evolved that formed white clouds when exposed to a glass rod wetted with ammonia solution.

(4)  When silver nitrate solution was added to the aqueous solution of the by-product, a white precipitate was formed.

(15) Line 140 eq. 1. It is not an equation. It is the mechanism scheme.

Our response: Done. Please refer to Line 154.

Scheme 1. Preparation of the trimellitic anhydride isothiocyanate cross linker.

(16) After crosslinking the amino functional group decreases in chitosan. please calculate how much % decrease. Authors can follow and cite this paper to calculate it. international journal of biological macromolecules 136 (2019) 661-667.

Our response: FTIR spectra (Figure 1) showed the evidence for the complete consumption of the amino groups of chitosan during crosslinking process. The doublet peak of the primary amine groups at 3352 and 3201 cm-1 has disappeared and completely replaced by a single peak at 3291cm-1 (H10), at 3296 cm-1 (H31), at 3288 cm-1 (H11) and at 3260 cm-1 (H13) related to the NH groups, confirming the reaction of all the amino groups of chitosan with the cross-linker. Please refer to Line 339-343. However, we cited your suggested paper to be applied as a general way to calculate the decrease in amino groups after crosslinking process (reference No. [11])

(17) Line 144-145 (5.0 %wt/v), please write correctly as 5 wt% v/v.

Our response: Done. Please refer to Line157-159.

(18) Line 145, how PVA dissolves in 60 °C? PVA melting point is higher than 60 °C. Please correct it.

Our response: PVA dissolves in distilled water at 60 °C to produce PVA solution. Please refer to Line157-158.

(19) Line 161, write correctly as et al.

Our response: Done. Please refer to Line 174.

(20) Line 154 to 155 please provide the optical images of hydrogels in the supplementary information.

Our response: Done.

(21) Why authors choose only H31 sample to prepare H31 /silver nanocomposites? Is there any specific reason please indicate it inside the manuscript?

Our response: Done (please refer to Line 565-568.

Although H31 is the most potent hydrogel in inhibition of the activity of the tested microbes in comparison to the other two hydrogels H11 and H13, its inhibition efficiency is lower than that of the used standard drugs Vancomycin and Amphotericin B. So, we selected H31 to prepare H31 /silver nanocomposites to enhance its antimicrobial activity. As would be expected H31/AgNPs5% and H31/AgNPs3% were more potent than Vancomycin and Amphotericin B against most of the tested microbes.

(22) Line 162, write mol L-1.

Our response: Done. Please refer to Line 176.

(23) Authors need to provide NMR data of cross-linked chitosan.

Our response: Unfortunately, we could not measure NMR spectrum of cross-linked chitosan because it is insoluble in all the commonly used solvent for NMR spectra measurement.

(24) line 206, the sentence should not start with a number (150 µL) please rewrite it.

Our response: Done. Please refer to Line 220.

(25) Line 222 please write clearly superscript.

Our response: Done. Please refer to Line 236.

(26) In 2.3.2 in FTIR section, please write the reference and scanning number.

Our response: Done. The scanning number is 16 (please refer to Line 263). For recording FTIR spectra, all the samples were used as powder without using KBr.

(27) Line 254, -7 please write in the correct way. Check in whole manuscript these types of errors.

Our response: Done.

(28) In all figures please clearly indicate major, minor ticks as well as x-axis and y- axis caption. All figures resolution is low.

Our response: Done.

(29) Merge figures 1 and 2 into one. And all the curves place into a stack format. 

Our response: It was difficult to merge Figures 1 and 2 together in one Figure because the intensity of the peaks became very weak and were not observable. Thus, it was better to merge 6 curves of chitosan PVA, H10, H11, H13 and H31 in Figure 1 and merge 3 curves of H31 and the AgNP composites in Figure 2. I hope you be convinced with that.

(30) make all the bands and its assignment into a tabular format for clear understanding.

Our response: Since we labelled all the characteristic peaks at all the curves in Figures 1 and 2 and explained their assignment in section 3.2.2. So, there is no need to tabulate them.

(31) Line 306-313 need to insert citations. Please insert it this line or other places in manuscript. Carbohydrate Polymers, 2022, 287, 119318, Carbohydrate Polymers 257 (2021) 117633 Carbohydrate Polymers, 2021, 261, 117875 and Cellulose, 29 (2022) 2399-2411.

Our response: Done (Please refer to Line 98-101 and Line 76). All your suggested references have been cited. Please refer to References No. [14], [23], [24], [25].

(32) Line 342. It is not an equation. It is a scheme. Please correct it accordingly other schemes.

Our response: Done. Please refer to Line 358.

(33) Figure 3 is messy please present it clearly. Deconvulted spectra of Ag and S need to provide. Other samples XPS data also need to provide.

Our response: In this work, we tried to provide further evidence for proving AgNPs formation into the matrices of H31 using XPS technique. We did not measure XPS spectra for sulphur because its presence was strongly proved by elemental analysis (Table 1). Unfortunately, we cannot improve Figure 3. The XPS spectra recorded for all the AgNP composites were similar. So, we provided XPS spectrum of H31/AgNPs5% composite as a representative example.

(34) Line 485-493 can be rewritten in a better way.

Our response: Done. Please refer to Line 487-501.

(35) Figure 4 is missing and merged into figure 5. Calculate all the sample's crystallinity data. Please follow and cite the following papers Journal of Luminoscense, 228 (2020) 117593.

Our response: It was difficult to merge Figures 4 and 5 together in one Figure because the intensity of the peaks became very weak and were not observable. Thus, it was better to merge 6 XRD patterns of chitosan PVA, H10, H11, H13 and H31 in Figure 4 and merge 2 XRD patterns of the AgNP composites in Figure 5. I hope you be convinced with that. The diffraction peaks were indexed to crystal planes of (111), (200), (220), (311) and (222) of face-centered cubic (fcc) Ag, which were consistent with that of Ag (JCPDS No. 4-0783). From the crystallinity data, the crystal structure of AgNPs was fcc (please refer to Line 409-411).

The suggested reference has been cited (Please refer to reference [39]).

(36) figure 5 y-axis title is missing.

Our response: Done.

(37) Figure 6 is missing. I am not sure why authors were not provided.

Our response: At initial submission All the Figures were provided in the word file of our manuscript before uploading it to the journal link. We are not sure some of these Figures might be lost during uploading.

(38) The authors have cited too many self-citations please reduce it.

Our response: Done. Reference [11], [14] have been omitted. Now our self-citations are only 5 references [10], [13], [30], [31] and [32].

(39) Figure 7 please indicate the scale bar clearly.

Our response: Done.

(40) In section 3.2.8, solubility behavior, what about others please provide the data if it swells or dissolves if dissolve calculates the exact amount. Additionally, when increasing the pH what is the effect of solubility in respective pH. please determine from UV data. Authors can follow this paper and cite Materials Letters 316 (2022) 132046.

Our response: All the prepared hydrogels and AgNP bio composites didn't dissolve in in a variety of solvents like water, 1% aqueous solution of acetic acid, N,N-dimethyl formamide, tetrahydrofuran, chloroform, ethanol, acetone and methylene chloride. since their initial weights were completely retained without any loss. Please refer to Line 463-468.

Thank you very much for your valuable suggestion. We will take this advice into our consideration in a separate study in the near future. We will study the swell ability of these hydrogels and AgNP bio composites at different temperatures and at a wide range of pH values in the presence of various salt concentrations. However, we cited your suggested paper to be applied as a general way to determine the solubility (Please refer to Reference [43]).

(41) Why PVA is not showing antibacterial and antifungal effects?

Our response: Because PVA is a synthetic vinyl polymer and it has a weakly acidic nature due to its hydroxyl groups. So, it cannot interact with the negatively charged membranes of bacteria and fungi. Please refer to Line 58-59.

(42) Why PVA/CS showing antibacterial and antifungal effects?

Our response: Due to the polycationic chitosan chains which are electrostatically interacted with the polyanionic bacterial cell membranes. This leads to a change in the permeability of these membranes and a loss in the internal electrolytes and proteineous ingredients of the bacterial cells [43]. Also, chitosan can amalgamate with DNA of the bacteria, restraining the synthesis of the protein and mRNA [44]. Further, the characteristic ligand ability of chitosan with metal salts, major nutritious materials, and spore elements affects greatly on the growth of the microbes [45]. Thus, incorporation of PVA with H10 in the same hydrogel improved the antimicrobial activity of PVA. Combination of PVA with chitosan, one of the antimicrobial natural biopolymers, is considered as an efficient approach to improve its antimicrobial activity and for further expanding of its application domains. Please refer to Line 514-541.

(43) Table 3, footnote, line 718, ml should be written as mL. please correct other places in the manuscript.

Our response: Done.

(44) Table 3 need statistical analysis. it is better to present it as a bar graph. Merge Table 3 and 4 into one bar graph.

Our response: The presented data in Table 3 is few to do statistical analysis. But we can describe the relation between the sample concentration and the cell viability is strong and inversive.

Please refer to Figure 13 (the merged Tables 3 and 4 into one bar graph)

(45) All figure charity so low. Author written figure but there are no figures 

Our response: We improved the clarity of all the Figures. We hope the word file of the revised manuscript upload without missing any Figure.

(46) The conclusions section needs to be shortened with important findings of this study.

Our response: Done.

(47) Check references font style. It seems different from then the journal guidelines and the main text of the manuscript.

Our response: We followed the journal guidelines during preparation of our manuscript.

Thank you very much for your valuable comments and suggestions. I hope that our responses are convincing to you and allow us to publish our revised manuscript in this distinguish journal, Polymers.

I am looking forward to hearing from you.

With our best regards

Mrs Reem T. Alfuraydi

Department of Chemistry

Faculty of Science

Qassim University

Reviewer 2 Report

The manuscript “Synthesis, characterization and evaluation of the antimicrobial and anti-biofilm formation activities of novel polyvinyl alcohol hydrogels reinforced with trimellitic anhydride isothiocyanate-crosslinked chitosan and loaded with silver nanoparticles” deals with the production of crosslinked blends of chitosan and PVA loaded with silver nanoparticles, characterized by an antimicrobial activity against various strains of Gram-positive bacteria, Gram-negative bacteria and fungi. The work is accurate and well organized; therefore, it deserves the publication. However, some revisions are required, as follows:

- Abstract. This section is very similar to the last part of the Introduction. Rewrite.

- Introduction. The production of PVA porous materials, loaded with silver nanoparticles, can be enlarged, adding to the state of the art alternative and innovative production processes, as the ones described in Baldino et al., Production, characterization and testing of antibacterial PVA membranes loaded with HA-Ag3PO4 nanoparticles, produced by SC-CO2 phase inversion, Journal of Chemical Technology and Biotechnology, 2019, 94(1), pp. 98–108; Prihandana et al., Study Effect of nAg Particle Size on the Properties and Antibacterial Characteristics of Polysulfone Membranes, Nanomaterials, 2022, 12(3), 388; Wang et al., Preparation and Properties of Silver-Based Cellulose/Polyvinyl Alcohol Antibacterial Materials, Journal of Inorganic and Organometallic Polymers and Materials, 2020, 30(11), 4382-4393; etc..

- Conclusions are a summary of the work. Rewrite in a more critical way, highlighting the main findings of the work.

- Correct typos.

Author Response

Comments and Suggestions for Authors

The manuscript “Synthesis, characterization and evaluation of the antimicrobial and anti-biofilm formation activities of novel polyvinyl alcohol hydrogels reinforced with trimellitic anhydride isothiocyanate-crosslinked chitosan and loaded with silver nanoparticles” deals with the production of crosslinked blends of chitosan and PVA loaded with silver nanoparticles, characterized by an antimicrobial activity against various strains of Gram-positive bacteria, Gram-negative bacteria and fungi. The work is accurate and well organized; therefore, it deserves the publication. However, some revisions are required, as follows:

(1) Abstract. This section is very similar to the last part of the Introduction. Rewrite.

Our response: Done. Please refer to Line 121-133.

(2) Introduction. The production of PVA porous materials, loaded with silver nanoparticles, can be enlarged, adding to the state of the art alternative and innovative production processes, as the ones described in Baldino et al., Production, characterization and testing of antibacterial PVA membranes loaded with HA-Ag3PO4 nanoparticles, produced by SC-CO2 phase inversion, Journal of Chemical Technology and Biotechnology, 2019, 94(1), pp. 98–108; Prihandana et al., Study Effect of nAg Particle Size on the Properties and Antibacterial Characteristics of Polysulfone Membranes, Nanomaterials, 2022, 12(3), 388; Wang et al., Preparation and Properties of Silver-Based Cellulose/Polyvinyl Alcohol Antibacterial Materials, Journal of Inorganic and Organometallic Polymers and Materials, 2020, 30(11), 4382-4393; etc..

Our response: Done (Line 92-94 and Line 103). Please refer to references No. [21], [22] and [26].

(3) Conclusions are a summary of the work. Rewrite in a more critical way, highlighting the main findings of the work.

Our response: Done.

(4) Correct typos.

Our response: Done.

Thank you very much for your valuable comments and suggestions. I hope that our responses are convincing to you and allow us to publish our revised manuscript in this distinguish journal, Polymers.

I am looking forward to hearing from you.

With our best regards

Mrs Reem T. Alfuraydi

Department of Chemistry

Faculty of Science

Qassim University

Round 2

Reviewer 1 Report

Authors revised their manuscript carefully and it is satisfactory. Now manuscript can accpet to publish in "Polymers" journal.